# Exploring the Mental Health Challenges and Coping Behaviour of Lesbian, Gay, and Bisexual Students at an Institution of Higher Learning

**DOI:** 10.3390/ijerph20054420

**Published:** 2023-03-01

**Authors:** Gsakani Olivia Sumbane, Nogwane Maureen Makua

**Affiliations:** Faculty of Health Science, School of Medicine, University of Limpopo, Private Bag X 1106, Sovenga 0727, South Africa

**Keywords:** lesbians, gays, bisexuals, mental health, coping behaviour

## Abstract

The South African university community is predominantly heterosexual, which fosters stigmatisation and discrimination against LGBTQI students despite the efforts to create conditions where LGBTQI students can succeed academically, socially, and personally. The study aimed to explore and describe the challenges experienced by LGBTQI students and their mental well-being as well as the coping behaviours adopted in a university in South Africa. This was accomplished using a descriptive phenomenological approach. A snowballing sampling method was used to select ten students who identified themselves as gay, lesbian, and bisexual (LGB). Semi-structured one-on-one interviews were conducted, and data were analysed thematically. The students perceived character defects stigma from fellow students and lecturers in and out of class. The mental health challenges experienced included a diminished sense of safety, lack of a sense of belonging, low self-esteem, and acting out of character. As a result, confrontation, passive withdrawal, and active dependent behaviour were utilised as different types of coping behaviour. The LGB students were subjected to stigma that negatively affected their mental health. Therefore, creating awareness about the rights of LGBTQI students to education, safety, and self-determination is recommended.

## 1. Introduction

South African University or campus communities are reported to be mainly heterosexual, which encourages a lot of prejudice towards lesbian, gay, bisexual, transgender, queer, and intersex (LGBTQI) students, according to Brink [1] and Gnan [2]. The heteronormative environment in the institutions of higher learning was not only reported in South Africa but also in the Southern African Development Community (SADC) as highlighted by Nduna, et al. [3]. The previous research on South African universities by Bhana [4], McLachlan [5], Sithole [6], and Copp, et al. [7] showed that students who belong to the LGBTQI community experience numerous challenges on campuses. These include verbal harassment, physical attacks, cyberbullying, social isolation, rejection, increasing feelings of insecurity and experiences of victimisation, and theft or damage to their personal property by other students, as reported by Gonzalez, et al. [8], Shramko, et al. [9], and Semlyen, et al. [10].

Nyeck et al. [11] indicated that the heteronormative environment in South Africa may be due to a view that same-sex relationships are “not African” but rather are seen as a Western creation. Similarly, in Taiwan, they contended that homosexuality is unnatural and deviant and that legalizing same-sex partnerships will promote homosexuality in society, create major epidemics of HIV infections, and depopulate the country, as reported by Lin et al. [12]. There is also evidence that South African society thinks homosexuality is a condition that can be “treated” or “fixed”, as highlighted by De Ru [13] and Mashabane [14]. One explanation, according to De Ru [13], could be that part of South African society’s fundamental ideas, which shape its social standards and values, are drawn from the Christian church. Even though South Africa’s legal system is among the most progressive in the world, particularly when it comes to the rights of sexual minorities, local research shows that social attitudes and norms are not always in line with the country’s progressive legislation (Kleinhans [15]; Nyeck et al. [11]). Instead, many South Africans continue to support monogamy and heterosexuality ideas. Additionally, Leburu et al. [16] claimed that similar concepts and attitudes permeate tertiary educational institutions, putting homosexual students in danger of discrimination or being subjected to violence. Therefore, LGBTQI students are more likely to experience mental health problems due to the difficulties encountered at tertiary institutions than their heterosexual colleagues, according to Gnan [2].

Lesbian, gay, bisexual, transgender, queer, intersex, and asexual individuals are referred to by the acronyms LGB, LGBT+, LGBTQ, LGBTQI, and LGBTQIA+. The abbreviations LGB and LGBTQI will be used in this article. The term LGB will refer to the participants, while LGBTQI refers to all individuals of this minority group in this article.

### 1.1. Current Incidents of Homophobic Attacks in South African Schools

Naude [17] and Zagagana [18] described South African schools as a hostile environment to LGBTQI and gender nonconforming students for too long, citing the following incidents as published by Roxburgh [19] from the *Mail & Guardian* news: the Soweto boy committed suicide on August 2022, claiming that a student teacher had humiliated him because of his sexuality; a 14-year-old student in the Eastern Cape also killed himself in August after being ridiculed at school. Similarly, another teen committed suicide in June 2022 as a result of ongoing bullying at school. Other incidents that took place in 2020 include a 16-year-old who was stabbed 13 times and killed in Khayelitsha because of his sexual orientation; a teacher in Robertson was attacked by parents because of his sexual orientation; and a visiting pastor in Cape Town warned students that, homosexual people are going to hell, as stated by Naude [17] from the African News Agency.

### 1.2. Brief Overview of the University Experiences Amongst South African LGBTQI Students

Higher education institutions are concerned about the creation of institutions free from prejudice, as reported by Munyuki et al. [20]. However, Jones [21] and Nduna et al. [3] showed that stigmatisation and discrimination of homosexual people at higher education institutions continue to be a challenge.

A study on the experiences of lesbian students by Leonard [22] revealed that the participants reported encountering homophobic behaviour experiences including psychological trauma, exclusion from social situations, and verbal, physical, and/or sexual abuse. However, other researchers believed that homophobic practices in South African institutions of higher learning are influenced by African cultural interpretations and selective readings of religious texts (Nduna et al. [3]).

Another study conducted with university students by Sithole [6] found that the LBTQI group of students are labelled as “sissyboys”, “Isitabane”, or “trassis”, which are terms for homosexuality in South Africa. Overall, 67% of the participants reported feeling excluded from class discussions by the lecturers and other students, whereas 58% of participants thought they received unfair treatment at the campus medical facility. Healthcare providers on-campus support discrimination by refusing to treat students. Discrimination was also noticed at the residence.

### 1.3. Brief Overview of Measures to Combat Homophobic Practices in South African Institutions of Higher Learning

Renn [23] revealed evidence that universities are creating conditions where LGBT students can succeed academically, socially, and personally. Munyuki et al. [20] highlighted that South African universities have put in place a range of formal policies that express their commitment to creating an environment for students that is free of discrimination, and many of these policies explicitly mention sexual orientation. Different programmes focusing on the LGBTI community on campuses are conducted. These include creating Gender Equity Units, planning Gay Pride marches, symposiums, campaigns, and closing functions at the end of the year as well as establishing support groups and supporting the annual International Day Against Homophobia, Biphobia, and Transphobia (IDAHOBIT) (Brink [1]; Zagagana [18]; Makapela [24]; Sefali [25]).

In addition, the provincial Department of Education (WCED) drafted the country’s first Gender Identity and Sexual Orientation Guidelines according to Naude [17]. The Western Cape Department of Education’s proposed guidelines is a crucial first step in creating a school atmosphere that is tolerant of gender and sexual diversity. The guidelines aim to sensitise public schools and the education environment to LGBTQI rights, in the promotion of a more inclusive approach and progressive realisation of Constitutional rights.

### 1.4. Impact of Continuous Exposure to Homophobic and Transphobic Attacks on University Students

The general mental and social well-being of LGBTI students are negatively impacted by their constant exposure to homophobic and transphobic attacks (Jones [21]). Some turn to suicide, while others experience emotional distress and alcohol and drug abuse (Moagi, et al. [26]. Renn [23]; Gnan [2]). Their academic performance suffers, and the ongoing stigma increases the number of students who drop out of school (Kosciw et al. [27]; Sukamti et al. [28]). According to the Mental Health Foundation (2021) in the UK, this particular group is more likely than heterosexual students to have poor mental health.

Yet, there is very little empirical research in the nursing science literature on issues related to the discrimination and stigmatisation of LGBTQI students at higher education institutions, as well as the impact on their mental health. There is virtually no attention on issues related to LGBTIQ students in nursing education. In addition, no study of this type has ever been undertaken in South Africa. Thus, the researchers decided to conduct this study in a nursing department at a university. This study is designed to address this gap and identify some key issues for improving the mental well-being of these particular students on campus.

### 1.5. Theoretical Framework

Erving Goffman’s [29] theories of stigma and labelling, as well as Link and Phelan’s [30] conceptualization of stigma, guided this study. The stigma theories were used to investigate the types of stigmas experienced by LGBTQI students at a university, as well as the effects of the stigma on their mental health and how they deal with their mental health challenges. Furthermore, the theories influenced how the findings were interpreted. Erving Goffman [29] defined stigma as an “attribute that is deeply discrediting”. A discredited attribute could be obvious, such as race, nation, or religion, or it could be hidden but still discreditable if revealed, such as sex workers, homosexuality, or a criminal record. Goffman [29] described stigmatised people as those who do not have full social acceptance and constantly strive to adjust their social identities.

Stigmatisation is reported to often go hand-in-hand with labels of deviance according to Goffman’s [29] theory of labelling. Goffman [29] defines labelling as a process of how labels are constructed and applied to certain individuals or groups to curtail or denigrate their actions. Stigma is about applying a label to describe someone’s perceived (or otherwise) non-conformist, degenerate, or simply different behaviour.

Link and Phelan [30] highlighted that once a person is labelled, there is very little that person can do to shake off this label. Clair [31], a social psychologist, noted that stigmatisation has been linked to several negative outcomes, including harm to one’s self-worth, academic performance, mental health, and physical well-being. Stigmatised individuals reported managing their stigmatised identities and coping with specific instances of discrimination by adopting numerous coping responses to stress such as avoidance, suppression, and identity development (Drapalski [32]; Patel & Hanlon [33]).

### 1.6. Research Aim

The researchers considered that performing this study on LGBTQI students was critical because the South African community is predominantly heterosexual, which fosters prejudice against LGBTQI students. The researchers believed that the problems associated with discrimination could hurt the mental health, academic performance, and success of students. Regardless of societal discrimination and stigma based on sexual orientation, a student must be emotionally stable to observe, think critically, and make appropriate decisions both in and out of the classroom. Therefore, the results could then be used to help guide intervention efforts on the part of the University when seeking to create a more welcoming environment for LGBTQI students. This study, therefore, explored and described the challenges experienced by LGBTQI students that might affect their mental well-being and the kinds of coping behaviour adopted when dealing with those challenges.

## 2. Research Design and Method

A descriptive phenomenological approach was adopted for this study. Employing this approach, the researchers were able to explore and describe the rich, in-depth lived experiences and coping behaviour of the selected students. The students were encouraged to link their day-to-day experiences to the challenges that might affect their mental health and well-being and different kinds of coping behaviour. Using this design, the researchers were able to describe the meaning of LGBTQI students’ actions and behaviours.

### 2.1. Research Setting

This study was conducted at the School of Health Care Sciences, Nursing Science Department in one of the selected institutions of higher learning in Limpopo Province, South Africa. At the time of this study, the Nursing Science Department offered undergraduate, post-basic, and postgraduate courses which include a four-year undergraduate bachelor’s degree in Nursing Science (R425), a one-year postgraduate diploma in Clinical Nursing Health Assessment, Treatment and Care (R48), and master’s and doctoral degrees. The setting was selected because LGBTQI undergraduate students were becoming more visible throughout the campus and exploring their mental health issues would help to raise awareness of their mental health problems and how this minority group could be accommodated and supported.

### 2.2. Population of Interest and Sampling Strategy

The first two participants were known to the researcher as LGBTQI, and they self-disclose their sexual identity. Thereafter, the other participants were identified using a snowballing sampling technique. Snowball sampling is a technique for locating people who would otherwise be difficult or impossible to locate according to Creswell [34]. The first two participants referred the researcher to seven participants who self-identify as LGBTQI. Of the seven, two declined to participate in the study. The researchers were referred to an additional three participants.

Although the sample size seems to be small, snowball sampling was used because the sample size was not easily identified in an institution of higher learning similar to where this study was conducted. The rural context and culture and stigma define how this population live. The sample size in phenomenological inquiry is relatively small because the aim is to acquire information that is useful for understanding the complexity or context surrounding the phenomenon. Methodologists such as Creswell [34] recommend a sample size of 3–10 participants for a phenomenological inquiry.

#### 2.2.1. Inclusion Criteria

This study included all those who had expressed an interest in participating, and who self-identified as LGBTQI in the second, third, and fourth year of study because they had lived experiences regarding the phenomenon on campus.

#### 2.2.2. Exclusion Criteria

This study excluded participants who declined to participate and those in their first year of study as they had less experience. Table 1 below illustrates the demographic data about the participants.

Four participants between the ages of 19 and 23 years identified themselves as gay, four as lesbians, and two as bisexual. Three participants were at level 2, four at level 3, and three at level 4, as illustrated in Table 1.

### 2.3. Data Collection

Data were gathered using semi-structured face-to-face interviews. Data were collected in a private room in the nursing science department skills laboratory after school when there was no disruption, over the course of two months. The researchers told the participants that audio-recording equipment would be used before the semi-structured interviews began. Participants were guaranteed confidentiality and given the choice to withdraw from this study if they felt uncomfortable participating in this study. The following questions were posed to all participants; however, other probing questions were dependent on the participant’s response.

Please describe in detail the challenges you typically face on campus that are affecting your mental health as an LGBTQI student.Describe fully the coping strategies you use to deal with your problems.What should the university do to establish a conducive environment for all students?

Data were collected up until the point of saturation during the eight interviews. However, two further interviews were undertaken to confirm the data saturation.

### 2.4. Data Analysis

Participants’ descriptions of the challenges they faced with their mental health and the kinds of coping behaviour that they employed to deal with the challenges at the higher education institution were analysed using the thematic approach by Nowell, et al. [1]. The researchers reviewed the transcripts while listening to the recordings and rectified any spelling or other problems in the transcripts. The authors independently coded the data by going over participant responses. Based on related concepts or meanings, all of the responses were sorted and put together. Following the analysis of participant responses, words or sentences were coded and grouped. These groupings created the initial themes. Then, the authors met to decide on the initial themes. An auditor reviewed the preliminary themes and gave the authors feedback. After carefully going over the data and coding quotes in accordance with the themes, the authors modified the thematic structure and constructed themes for the dataset. A consensus meeting was set up between the researchers to address any differences in opinion that cropped up during the coding process, to update the initial themes, to find new topics, and to eliminate themes until the final themes and subthemes were produced. Thus, themes that directly pertained to and answered the research question were identified. These themes are presented in the results set out below.

### 2.5. Ethical Considerations

This study was submitted to the Turfloop Research and Ethics Committee for ethical clearance and approval (TREC/497/2019: PG). Permission to undertake the study was obtained from the Director of the Health Care Sciences School as well as the Head of the Nursing Science Department. Individual participants in this study gave their informed consent. Participants were given complete information about this study, including the objective, risks, and benefits, what was expected of them, and that confidentiality would be maintained so that they could give informed permission. They were also informed about the data collection tool. Confidentiality was ensured because each participant was given a code name, and these codes were used when discussing data, as recommended by Creswell [34]. The participants were not required to identify themselves by name during the recording. The name of the institution at which this study was conducted is also not mentioned in this report. Furthermore, the information that had been obtained was kept in a secure location to which only the researchers had access. The information was kept under lock and key and only the researchers had access to the key. The questions put to the participants were not sensitive and could not cause emotional harm.

## 3. Results

Three themes and nine sub-themes emerged during the analysis of the data collected. The themes and sub-themes that identified the mental health problems and coping behaviour of LGB students at the selected institution of higher learning are presented in Table 2 below.

During the period of the interviews, the researchers noted that some of the participants seemed relieved to have found someone who could listen to their distressing life stories. They were freely expressing their challenges, and at times, their eyes filled with tears revealing their feelings of helplessness and hopelessness when they described the emotions that they experienced relating to the circumstances affecting their mental health. These circumstances included bullying, teasing, and verbal abuse. However, when they described their coping mechanisms, they smiled and laughed to show that they had eventually developed a tolerance for internal and external distress. They seemed cautious about sharing their experiences, as shown by the fact that they were a bit reticent when talking about their sexual identity.

### 3.1. Theme 1: LGB Students’ Mental Health Challenges

Theme 1 described the challenges experienced by the LGB students on campus that are affecting the mental health of LGB students on campus. These include a diminished sense of safety, low self-esteem, a loss of a sense of belonging, and acting out behaviour were the most common mental health challenges affecting the mental health of LGB students.

#### 3.1.1. Sub-Theme 1.1 Diminished Sense of Safety

This study revealed that LGB students perceived bullying, teasing, labelling, verbal abuse, and lack of privacy from fellow students and lecturers in and out of the class. The participants were found to experience stigmatisation and discrimination on campus based on their character. As a result of the stigmatisation and discrimination that the LGB students were subjected to, their sense of safety was found to be diminished, which negatively affected their mental health.

As reported by gay participants in the following excerpts:


*One day, we were attending a bash on the campus and this other guy was passing by and accidentally broke my beer bottle. Instead of apologizing because it was a mistake, he said to me that he wouldn’t apologize because I am a stabane [Gay] and if I decide to fight him, he will take off my dirty pants and show the people that I have male private parts but I decided to squeeze them in between my thighs. [sigh]. “He went on saying I’m wasting God’s gifts by misusing my private parts”… [Eyes filled with tears]*
(22-year-old, third-year student identified as gay). 


*When I first came to the campus, I was bullied, especially by my resident mates, who are male residents. They used to tease me, telling me that I was lost and should go to the female resident to stay with the other girls because I act like a girl*
(20-year-old, second-year learner, identified as gay). 


*One day, a lecturer pointed at me and said, “You’re always sitting at the front desks but you never answer. This question is yours to answer”. Then he asked me a question. Unfortunately, I didn’t know the answer, so I said to the lecturer, “Sir, I sit in front because I have hearing problems”. I then continued and told him I didn’t know the answer. One guy was seated behind me. He shouted, and I said, “Sir, that boy is sitting in front of you to seduce you. Can’t you see?” He laughed*
(22-year-old, 4th-year student identified as gay). 

The frustration expressed by the LGB students is a reflection of the stigmatisation that has affected not only their physical and mental well-being but also their material resources. As evidenced in the following excerpts:


*My roommate used to search my wardrobe when I was away. I don’t know what he was looking for, but one day I caught him searching and I asked him what he was looking for. He told me that he wanted to see if I was wearing female panties or male jockeys because he didn’t understand me. That day I felt so frustrated and felt like packing my bags and quitting school.*
(22-year-old, 4th-year student identified as bisexual).

Other participants reported that their roommates were anxious and uncomfortable about sharing a room with them because they did not feel comfortable undressing in front of them. Some were afraid that the LGB students would rape them, and some felt safe when they had a visitor, as evidenced by the excerpt below:


*My roommate would always tell me that I was making her uncomfortable when we started living together. She used to ask me to excuse her when she had to undress, and she would also excuse herself when she saw me undressing. She said it makes her uncomfortable because she looks at me as a male, not a female. I tried to explain to her that I am also a girl, just a lesbian, and I wouldn’t do any harm to her, but obviously, it took her months to get used to me. She struggled a lot, and I felt very sorry for sharing a room with her, but unfortunately, there was nothing I could do.*
(21-year-old, 3rd-year student identified as lesbian). 


*My roommate used to tell me that I was making her uncomfortable in her room because I wasn’t trusting her. She said she was not sure if I wouldn’t rape her as I am a lesbian.*
(20-year-old, 3rd-year student identified as lesbian). 


*I have realized that my rumza [roommate] is uncomfortable when there are only the two of us in the room, but when one of us has a visitor, she is happy and free. Thus, the thing of sharing rooms doesn’t work, and I don’t want to lie*
(19-year-old, 2nd-year student identified as lesbian). 

#### 3.1.2. Sub-Theme 1.2: Lack of a Sense of Belonging

Participants mentioned how their colleagues on campus make them feel as if they are not a part of them because of their comments and strange looks, indicating that they are not accepted. As indicated by the following excerpt:


*My classmates used to look at me funny when we first started with our degrees. I could notice the weird looks because we were a class of a few students. Lol (laughing)*
(3rd-year students identified as gay). 


*…once I had an awkward conversation with a group of friends from class discussing sexuality and they mentioned that being gay is a choice. And I felt like I don’t belong there as they viewed me as my own decision*
(4th-year student identified as gay) 


*Sometimes people make you feel like an outsider because they refer to me as so and so that one who’s gay but when they refer to other people they refer to them using their names only*
(22-year-old gay in 4th year). 

#### 3.1.3. Sub-Theme 1.3: Low Self-Esteem Due to Fear of Being Judged

A male student who identified himself as gay remarked that most of the time it is difficult to participate in class due to his fear of being teased by classmates. He further stated that he would just keep quiet when the lecturers were asking questions, even though he wanted to attempt to answer, but he was scared. One participant added that he had also experienced teasing at the high school level. LGB students reported that class participation is a challenge as is shown in the excerpts below.


*At first, when I got here, I was afraid to participate in class. I would just keep quiet even though I knew the answers. I was afraid that if I happened to get the answer wrong, my fellow students would laugh at me as they used to at high school…*
(21-year-old gay in the 2nd year of study). 


*Sometimes we would be given schoolwork to work on as a group and I would just stand there watching everybody choose a group they wanted to be in. For me, I would find it difficult to choose a group because I thought, what if the group I chose doesn’t want me? Then I would wait for everybody to get a group, then wait for the last group that is short on members, and then I would join them. That way, I know they won’t refuse to pair with me as they didn’t have a choice.*
(22-year-old lesbian in the 4th year of study). 

#### 3.1.4. Sub-Theme 1.4: Acting out Behaviour

One of the students who identified herself as lesbian used to cry out loud during lectures in the classroom. Her actions had nothing to do with what the lecturer or any other students had stated in class. When the lecturer was in the middle of teaching, she would suddenly start crying, rush to the restroom and scream aloud. The majority of her lecturers made an effort to speak with her about the behaviour and even referred her for emotional help but the behaviour persisted. She stated that the following was the primary cause of the behaviour:


*I saw my partner with the security guard the other day; despite my prohibition on her having a social media account, she is always on it. I was checking her secretly the other day to see who she had been chatting with. I always inquired about her relationship with the security guard, but she always denied it, even though I was aware that something was going on between them. She doesn’t answer her phone when I’m around, and I’m worried about whom the partner is talking to on the phone or WhatsApp. This concerns me because we are in the same class and others may perceive me as a fool*
(21-year-old lesbian in the 3rd year of study). 

Another type of ‘acting-out’ behaviour reported was the consumption of too many substances, as expressed by one lesbian,


*“I drink too much alcohol with my male friends, almost every day because nursing is strenuous. We work very hard, we don’t even have time to rest”*
(20-year-old lesbian). 

### 3.2. Theme 2: Coping Behaviour

LGB students have been found to use various kinds of behaviour to attempt to relieve stress or threats related to the stigmatisation that negatively affects their mental health. These include passive withdrawal and, or active dependent behaviour, confrontation, and tolerance to internal and external distress.

#### 3.2.1. Sub-Theme 2.1: Passive Withdrawal Behaviour

The participants expressed that withdrawal or distancing themselves from the situation was one of the coping strategies that they utilised to cope on campus. To cope with the situation on campus, LGB students tend to utilise the following kinds of behaviour to avoid being bullied and teased by other students: going to sleep, missing classes, staying off campus, and having few friends, as evidenced by the following excerpt:


*He went on to say, “I’m wasting God’s gifts by misusing my private parts”. [eyes filled with tears]. “I immediately went to my room to sleep, and I slept very heavily that day”.*
(22-year-old gay). 


*“… teasing makes me skip classes sometimes”*
(21-year-old lesbian in the 4th year of study). 


*Since I don’t have many friends, I also told myself that it is not my fault that people choose not to understand homosexuality, so I am not going to allow them to stress me.*
(19-year-old gay in the 2nd year of study). 


*Because of the teasing, I ended up avoiding using the bathrooms at the specific residence. I’m now staying off campus to avoid having to share a room and a bathroom*
(a 20-year-old lesbian). 

#### 3.2.2. Sub-Theme 2.2: Active Dependent Behaviour

Participants were found to have developed ways of dealing with the stigmatisation and discrimination on campus. These include acceptance of the situation and coming to terms with it, focusing more on studies, getting support from lecturers, and joining a university LGBTQI support group, as shown in the following excerpts:


*Accepting and acknowledging that not everybody will understand your sexuality is working for me. I realized that I couldn’t make people understand and accept me as I am. They will understand when they are ready to replace ‘I don’t care anymore’*
(22-year-old gay in the 3rd year of study). 


*The thing is, we are not the same. Myself, when I feel stressed, I read my books and concentrate on my studies. I always tell myself that I have to pass my degree so that I can go and work and meet people with a better understanding of homosexuality. So being a homosexual undergraduate never really affected my studies.*
(20-year-old gay in the 2nd year of study). 


*“…the lecturer didn’t like the joke and teasing because he chased that guy out of class and then made it clear that he won’t accept bullying from other students in his class, I felt so good that day to realize that few people understand the bullying we going through…”*
(21-year-old bisexual in the 4th year of study). 


*We currently have a student health centre where there is a structure for LGBTQI students on our campus. We meet as homosexual undergraduate students and talk about our problems and encourage those who are struggling to come out to do so and advise them on how to come out*
(19-year-old lesbian in the 2nd year of study). 

#### 3.2.3. Sub-Theme 2.3: Confrontation

Other participants reported using confrontation to challenge their colleagues and raise awareness of homosexuality in their classrooms and residences, as well as to release stress and threats. As evidenced by:


*One day, I developed the strong guts to stand in front of them while we were still waiting for our lecturer to arrive. I tried to explain everything I could remember about homosexuality. They even asked some funny questions that I was able to answer, and after that conversation, they became my support structure for shame. We are now best buddies and everybody likes me so much. I am so grateful.*
(23-year-old bisexual in the 4th year). 


*Sometimes I would try to talk to my roommate to make her understand that I am just a girl like her; it is just that I am a lesbian. And like other girls, I don’t have feelings for everyone I meet; I have my types as well. I was trying to make her feel free and take out the mentality that maybe I wanted her (laughing)…*
(21-year-old lesbian in the 3rd year of study). 

#### 3.2.4. Sub-Theme 2.4: Tolerance of Internal and External Distress

The study revealed that to cope with their identity, some participants were shown to have the ability or desire to tolerate the existence of beliefs or kinds of behaviour that they disliked or about which they disagreed. This is illustrated in the excerpt below.


*I don’t expect everyone to understand me as I understand myself. I know being a lesbian is found to be unacceptable by some churches and their people. That doesn’t bother me a lot. I am happy with who I am. I cannot change my sexuality because people choose to not understand it.*
(21-year-old lesbian in the 4th year of study). 


*I am gay, and when people see me talking or walking with a guy they think I’m on that guy. Even the straight guys when you talk to them they think you are on them, so I am used to their attitude and I don’t care anymore.*


### 3.3. Theme 3: Suggested Forms of Support for LGB Students on Campus

The majority of LGB students stated that stigmatisation is so common that sometimes they find it difficult to deal with. They further remarked that being able to consult a professional counsellor every time they felt down would be beneficial to them. However, the majority of the participants pointed out that the university is not supportive enough when it comes to homosexuality issues. The following are some suggested forms of support that could be introduced for LGB students:

#### Sub-Theme 3.1: The Need for an Environment That Offers Opportunities for Recreation, Socialisation, Psychotherapy, and Safety


*“We need functions like Gay Pride where we meet as gays and enjoy ourselves together. This usually encourages other gay guys who are hiding to come out. It also helps the community to be aware that we exist and that we are many, so they can accept us if they are aware of that, I guess.”*
(22-year-old gay). 


*I just think that if the university hired some professional people to help us every time we experienced psychological problems, then I would say the university is trying its best. Right now, if you have problems as an LGBTQI, you have to deal with them by yourself or seek help from friends. But it would be nice to have psychologists around for proper professional help and guidance. These structures that we already have just need psychologists to be completed. For me, they are useless without professionals to help us.*
(22-year-old bisexual in the 4th year of study). 


*Maybe those professionals can develop strategies to reduce bullying and discrimination and implement them. That way, other students can learn and be aware of the things they are doing to us and realize that they are not right, and who knows, maybe the abuse will decrease*
(19-year-old gay). 


*Single rooms are a solution, according to me. Even in the first and second years, we must be given single rooms. At least in the final year, we are given single rooms, and it is so nice because you are also free to live your life without being judged by your roommate*
(a 21-year-old lesbian in the 4th year of study). 

## 4. Discussion

This research explored and described the challenges experienced by LGB students and their mental well-being as well as the coping behaviours adopted in a university in South Africa. The study found that LGB students perceived character defects stigma from fellow students and lecturers in and out of class. These include victimisation, bullying, labelling as “stabane”, funny looks, threats, insults, and teasing to belittle them on university premises (such as public spaces, bathrooms, and classrooms), which were based on their character. The findings are in line with Letsoalo [35], Sava, et al. [36], Antebi-Gruszka, et al. [37], and Treharne et al. [38], which reported similar findings.

The character defects stigma subjected to LGB students negatively affected their mental health. Therefore, the mental health challenges experienced included a diminished sense of safety and feeling of being unaccepted and unliked by others, resulting in a lack of the sense of belonging. Participants also reported low self-esteem, which is manifested by difficulties in choosing, joining, and participating in a discussion group because of a fear of how the other members would react. Acting out behaviour was also found to be a mental health challenge, whereby some participants were seen crying or screaming in class, and others reported to be drinking too much alcohol, which might predispose them to some mental health conditions such as alcohol dependency or substance-induced psychosis. A study conducted by Dau et al. [39] showed that over 25% of LGBTQI people at a university in Australia experience alcoholism, and there is a strong correlation between victimisation and alcohol usage. Another study conducted in Lesotho by Fantus et al. [40] showed that alcohol usage is a stigma-coping method as well as a way to create LGBTQI friendships.

Participants reported managing their stigmatised identities and coping with specific instances of discrimination by adopting numerous coping behaviours. First among these is passive withdrawal behaviour (disengagement by bunking classes, refraining from using the residence bathrooms, and staying off campus). Passive withdrawal behaviour was described as an unhealthy coping mechanism by Gnan [2], which has a negative influence on the mental health of this population. The second is active dependent behaviour (self-acceptance, concentrating more on schoolwork, and support systems such as friends, lecturers, and support groups) and confrontation. These kinds of behaviour were reported to be associated with better psychosocial adjustment and a greater likelihood that participants would attain their degrees by Goldbach, et al. [41] and Toomey, et al. [42]. The last strategy was tolerance, which involves changing the way they think about the experiences of LGBTQI stigma. Other suggested coping strategies that are required on campus include the gender-based unit, the gay pride march, and single rooms for students. Brink [1] and Zagagana [18] reported that to date, the majority of local and international institutions have a gender-based office and hold an annual gay pride march to create awareness and educate the university community about sexual orientation.

The findings of this study integrated with the theories of stigma since participants perceived character defects stigmatisation and labelling on campus. The stigmatised persons who are LGBTQI students exhibit a variety of negative outcomes such as low self-esteem, a loss of the sense of belonging, acting out behaviour, and an avoidant coping technique.

## 5. Conclusions

Given the fact that character defects stigmatisation was discovered to be the most significant challenge subjecting LGBTQI students to mental health challenges on campus, understanding the factors influencing heterogeneous students’ and lecturers’ behaviour among this group is a critical area of future research. The university should consider creating awareness about the rights of LGBTQI students to education, safety, and self-determination. Leaders of the LGBTQI support system could seek a time slot on the University’s radio station to discuss homosexuality-related issues and concerns. School health outreach should be organised by the Nursing Science Department to make high school learners aware of the effects of bullying and negative attitudes towards LGBTQI students. The university could develop a policy that protects the rights of this group of students and outlines the actions that would be taken if those rights are violated. Creating a consistent, ‘high-touch’ environment marked by safety, security, and regular communication is essential. These aims can be accomplished by allocating a residence to this gender minority group, allowing them the freedom to use the restrooms without worrying about being bullied. Fostering trusting relationships between learners, teachers, staff, and other individuals in programmes is also important. Organising a Gay pride march for this particular community once a year at which they can all meet and enjoy themselves without being interrupted by other students is a positive step. All LGBTQI students should be encouraged to use the student counselling centre to deal with the social and emotional issues that they face on campus.

The policymakers in South Africa should consider the drafted country’s first Gender Identity and Sexual Orientation Policy Guidelines, as it addresses most of the interventions for the challenges experienced by LGBTQI students, which include allowing for unisex bathroom facilities, consideration for sleeping arrangements, and recommendations for safe spaces and social diversity associations at schools.

### Strength and Limitations

Theories of stigma were used to guide the process of this study and to make research findings meaningful. The study adopted a descriptive phenomenological design to obtain complete and accurate information by allowing LGBTQI students to describe their day-to-day experiences that affect their mental health and coping behaviour. A limited number of participants was utilised, which limits the generalisability of the findings. Despite the limitations, the aim and objectives were achieved.

## Figures and Tables

**Table 1 ijerph-20-04420-t001:** LGB students’ demographic information.

Participant Code	Level of Study	Age	Gender	Sexual Orientation
P1	3	22	Male	Gay
P2	3	21	Female	Lesbian
P3	2	19	Male	Gay
P4	2	20	Male	Gay
P5	2	19	Female	Lesbian
P6	4	22	Male	Bisexual
P7	4	23	Female	Bisexual
P8	4	22	Male	Gay
P9	3	20	Female	Lesbian
P10	3	21	Female	Lesbian

**Table 2 ijerph-20-04420-t002:** Theme and Sub-theme.

Theme	Sub-Theme
LGB students’ mental health challenges	1.1. Diminished sense of safety 1.2. Lack of sense of belonging 1.3. Low self-esteem due to fear of being judged 1.4. Acting out behaviour
2. LGB students’ coping behaviour	2.1. Passive withdrawal behaviour 2.2. Active dependant behaviour 2.3. Confrontation 2.4. Tolerance of internal and external distress
3. Suggested needs of LGB students on campus	3.1. The need for recreation, socialisation, psychotherapy and a safe environment

## Data Availability

The datasets that support the findings of this study are available on request from the corresponding author, G.O.S. Because the data contains information that could jeopardize the privacy of research participants, it is not publicly available.

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
