# Peer review of "Exploring the Mental Health Challenges and Coping Behaviour of Lesbian, Gay, and Bisexual Students at an Institution of Higher Learning"

_ijerph, 2023, doi:10.3390/ijerph20054420_

Round 1

Reviewer 1 Report

Exploring the Mental Health Challenges and Coping Behaviour of Lesbian, Gay, and Bisexual Learner Nurses at an Institution  of Higher Learning

The study examined the experience of 10 undergraduate nursing students who identified as gay or bisexual to identify mental health challenges as well as adaptive behaviours.

In general, I got the impression that this is a very important study, aimed at making the voice of a vulnerable population group heard, and that the study is structured in a systematic and convincing manner.

However, I was under the impression that there are several essential corrections that need to be made, as follows:

introduction

1) As part of the introduction, examples of studies from certain countries (such as Colombia, for example) are given. It is not clear why the writers chose certain countries over others?

2) More specifically - what are the data about South Africa?

It is necessary to describe the context - in South Africa - what is the attitude towards the research population? Is it different from other countries in relation to the world? why?

3) At the end of the introduction - please summarize what is known about studies with this population group within higher education institutions and place the current study in relation to the gaps that exist in the more general research field.

Theoretical framework

1) At the end of the introduction or at the beginning of the section of the theoretical framework, a justification must be provided regarding the theory that has been developed. Why is it suitable for research and its results?

2) More specifically, it seems that the authors used an overly general and simplistic theory. I would expect to incorporate any kind of critical theory into a manuscript of this type, for example a theory of identity and stigmatization processes\a theory of diversity\the conceptualization of processes of dealing with stigma.

Statement of the problem

1) This part is worded too generally and does not flow. It has a kind of repetition of the introduction.

Research aim

1) Please indicate the importance of this type of research within an academic space. How is it different from research in the field of workplace/other spaces?

Sample

1) I would have expected to see wording that more specifically explains why the sample here is so small. Although there is an explanation of the process within this section, it is too cumbersome. I would have expected clearer wording.

The part of data collecting and presenting the resolt is formulated in a very good way.

Discussion

I propose to include in the discussion any theoretical aspects that connect the results of the research and theoretical bodies of knowledge.

Interpretation of the findings based on theory?

I would like to thank the authors for the opportunity to review such an important study and encourage them to complete the necessary revisions so that it can be published

Author Response

Heading

Comment

Effected corrections

Please confirm which names should be kept.

Please utilise the full names:  Gsakani Olivia Sumbane and Nogwane Maureen Makua

Abstract

The abstract was also amended based on the effected corrections

1st reviewer’s comments

Introduction

1) As part of the introduction, examples of studies from certain countries (such as Colombia, for example) are given. It is not clear why the writers chose certain countries over others?

South African context regarding the matter was discussed in detail. The following subtopics were added on the introduction:

-          incidents of homophobic attacks at schools, 

-          Measures to combat homophobic attack at the universities,

-          Impact of continuous exposure to homophobic on students

2) More specifically - what are the data about South Africa?

It is necessary to describe the context - in South Africa - what is the attitude towards the research population? Is it different from other countries in relation to the world? why?

South African context regarding the topic has been described and compared with other countries.

The attitude towards the research population was described

3) At the end of the introduction - please summarize what is known about studies with this population group within higher education institutions and place the current study in relation to the gaps that exist in the more general research field.

A paragraph before the theoretical framework addressed this comment

Theoretical framework

1) At the end of the introduction or at the beginning of the section of the theoretical framework, a justification must be provided regarding the theory that has been developed. Why is it suitable for research and its results?

A justification is provided regarding the theory that has been used and why it is suitable for the research and results.

The last paragraph on the discussion of the findings indicates how the findings were integrated with the theory.

2) More specifically, it seems that the authors used an overly general and simplistic theory. I would expect to incorporate any kind of critical theory into a manuscript of this type, for example a theory of identity and stigmatization processes\a theory of diversity\the conceptualization of processes of dealing with stigma.

Erving Goffman's theories of stigma and labelling, as well as Link and Phelan's conceptualization of stigma, was used to guide the research.

Statement of the problem

1) This part is worded too generally and does not flow. It has a kind of repetition of the introduction.

To avoid the repetition, the statement of the problem was delete and more information regarding the background of the study was described on the introduction and research aim

Research aim

1) Please indicate the importance of this type of research within an academic space. How is it different from research in the field of workplace/other spaces?

The research aim was discussed in detail

Sample

1) I would have expected to see wording that more specifically explains why the sample here is so small. Although there is an explanation of the process within this section, it is too cumbersome. I would have expected clearer wording.

Sampling is described as suggested with an explanation why sampling is small.

Discussion

I propose to include in the discussion any theoretical aspects that connect the results of the research and theoretical bodies of knowledge.

Interpretation of the findings based on theory?

Discussion of the results was described logically and more information has been added. The theoretical aspects that connect the research and theoretical bodies of knowledge has been added as suggested.

Reviewer 2 Report

This manuscript presents finding from a small, qualitative study of “nurse learners”, specifically those who identify as lesbian, gay, or bisexual, and their mental health challenges within the context of their learning environment. Potential contributions of this work include the incorporation of Maslow’s work into discussion of minority stress and coping experiences among LGB groups; the focus on LGB students in South Africa; and the emphasis on nursing students’ experiences specifically. In terms of writing mechanics, it is well-written.

Having said that, on the whole, this paper does not cohere, and there are many areas that are underdeveloped theoretically and methodologically. Though the aforementioned areas could  be contributions, presently, none are sufficiently grounded in literature/theory or adequately explored or explained to be impactful contributions. Additionally, the benefit of qualitative inquiry is the rich and “thick” description it can often provide. Unfortunately, the findings presented here feel very “thin” and at times, the data provided does not seem to match or support the coding scheme.

Here are questions that are not clearly answered in the manuscript that require more detailed and thorough answers:

-       Why is an emphasis on “learner nurses” relevant to Public Health?

-       What value is there to studying sexual minority nurses specifically? That is, why/how do nursing students differ from other students in such a way as to make the exploration of mental health and coping experiences of sexual minority students specific to just nursing students?

-       How does Maslow’s theory speak to/converge with other theories and literatures related to mental health and coping, which is the focus of the study? According to whom does “sexual expression” fall under physiological needs, as originally described by Maslow?

Questions about Methods and Results

-       What qualitative approach was employed in this study? The “method” was a semi-structured interview, but there is no discussion of the larger methodological orientation the researchers relied upon.  

-       How were participants screened for the study, and what was the specified inclusion/exclusion criteria?

-       In the Intro, authors make a deliberate choice to focus primarily on sexual orientation/identity, not gender identity, i.e., through choosing to use the language “LGB” (bottom of page 1). There is no rationale for why this choice is made. Further, and very, very confusingly, the demographic table on page 4 lists both “gender” and gender identity. First, male and female are sexes, not genders. More confusing, six of the ten participants (the majority) have a sex and gender that do not align, suggesting the majority of the sample is comprised of trans or non-binary persons, which is inconsistent with how the study is framed, described and discussed.

-       The interview guide is insufficiently described. Please provide more detailed information about the questions and probes used in the interviews. Also discuss how/by whom the interview script was created and designed.

-       Please include a positionality statement

-       Please review the difference between anonymity and confidentiality (page 5, 215-220). A study is one or the other- not both.

Overall, in many instances, “codes” and their sub-themes are not adequately described and/or the quotations don’t seem to support the code. For instance, 3.1.1 and 3.1.2, authors have labeled both as “mental health challenges” though few of the quotes from the participants themselves are about their mental health, specifically or broadly.

The sub-theme of “obsessive jealousy” does not belong in the coding scheme at all. It does not fit, or make sense within either the coding structure, or the larger aim of the study.  

The Discussion section seems to be a mixed bag of theories and perspectives, with each paragraph introducing a new theory/theorist, rather than a cohesive contextualization of findings, tied back to Maslow. It is disjointed and while “mental health” appears to be a through line, there is not singular framework used to explain the study’s findings.

What does the phrase “gender community mean”? This study was about sexual identity, not gender identity.

Discussion of strengths and limitation is, at best, underdeveloped.  

Author Response

Heading

Comment

Effected corrections

Please confirm which names should be kept.

Please utilise the full names:  Gsakani Olivia Sumbane and Nogwane Maureen Makua

Abstract

The abstract was also amended based on the effected corrections

2nd reviewers comments

Overall comment

Having said that, on the whole, this paper does not cohere, and there are many areas that are underdeveloped theoretically and methodologically. Though the aforementioned areas could  be contributions, presently, none are sufficiently grounded in literature/theory or adequately explored or explained to be impactful contributions. Additionally, the benefit of qualitative inquiry is the rich and “thick” description it can often provide. Unfortunately, the findings presented here feel very “thin” and at times, the data provided does not seem to match or support the coding scheme.

All comments regarding the methodology, theory, aim of the study and findings were corrected as suggested

Aim

Why is an emphasis on “learner nurses” relevant to Public Health?

-       What value is there to studying sexual minority nurses specifically? That is, why/how do nursing students differ from other students in such a way as to make the exploration of mental health and coping experiences of sexual minority students specific to just nursing students?

This comment is addressed on the aim of the study

Theoretically framework

How does Maslow’s theory speak to/converge with other theories and literatures related to mental health and coping, which is the focus of the study? According to whom does “sexual expression” fall under physiological needs, as originally described by Maslow?

Maslow theory was changed to

Erving Goffman's theories of stigma and labelling, as well as Link and Phelan's conceptualization of stigma seem to be more relevant to the study

Methods

-       What qualitative approach was employed in this study? The “method” was a semi-structured interview, but there is no discussion of the larger methodological orientation the researchers relied upon.  

The phenomenological approach was added

Sampling

How were participants screened for the study, and what was the specified inclusion/exclusion criteria?

Additional information like screening of the participants was added.

Inclusion and exclusion criteria was also amended

In the Intro, authors make a deliberate choice to focus primarily on sexual orientation/identity, not gender identity, i.e., through choosing to use the language “LGB” (bottom of page 1). There is no rationale for why this choice is made. Further, and very, very confusingly, the demographic table on page 4 lists both “gender” and gender identity. First, male and female are sexes, not genders. More confusing, six of the ten participants (the majority) have a sex and gender that do not align, suggesting the majority of the sample is comprised of trans or non-binary persons, which is inconsistent with how the study is framed, described and discussed.

The gender identity column in table 1 was deleted as it was confusing the readers

Data collection

The interview guide is insufficiently described. Please provide more detailed information about the questions and probes used in the interviews. Also discuss how/by whom the interview script was created and designed.

The questions asked during the interviews were added as suggested

Please include a positionality statement

Positionality statement is added

Ethical considerations

Please review the difference between anonymity and confidentiality (page 5, 215-220). A study is one or the other- not both.

Confidentiality was used instead of both

Findings

Overall, in many instances, “codes” and their sub-themes are not adequately described and/or the quotations don’t seem to support the code. For instance, 3.1.1 and 3.1.2, authors have labeled both as “mental health challenges” though few of the quotes from the participants themselves are about their mental health, specifically or broadly.

The sub-theme of “obsessive jealousy” does not belong in the coding scheme at all. It does not fit, or make sense within either the coding structure, or the larger aim of the study.  

“as a mental health challenge” was deleted on subthemes 3.1.1 and 3.1.2.

The obsessive jealousy on subtheme 1.4 was deleted and the subtheme was clearly described

Additional quotations was added to the following subtheme:

-          Lack of sense of belonging

-          Tolerance of internal and external distress

Most of the statements on the presentation of the findings were rephrased.

The Discussion section seems to be a mixed bag of theories and perspectives, with each paragraph introducing a new theory/theorist, rather than a cohesive contextualization of findings, tied back to Maslow. It is disjointed and while “mental health” appears to be a through line, there is not singular framework used to explain the study’s findings.

What does the phrase “gender community mean”? This study was about sexual identity, not gender identity.

Discussion of the results was described logically and more information has been added.

Gender community and gender identity was deleted.

Highlighted text

We highlighted some duplicated parts with other already published materials. We would suggest to rephrase them while revising your paper.

All highlighted text was attended to.

Round 2

Reviewer 1 Report

I get the impression that the writers did a good job in addressing the issues that required revision. The manuscript is now ready for publication and has the potential to contribute to research, discourse and practice about the challenges of students with different and diverse gender identities in academic spaces

Author Response

There were no further comments from this reviewer, hence no document was uploaded

Reviewer 2 Report

I appreciate the time and effort the authors have devoted to revisions and addressing reviewers' comments. The introduction has been improved significantly. I have remaining concerns about how participants were recruited, and still am not convinced that a study of "nurse learners" is broadly applicable, or more specifically, that it is a good fit with the current journal. The complete replacement of the original theoretical framework with an entirely new framework is very concerning, and suggests that the study was not theoretically grounded in the first place. 

Author Response

Comments

Effected corrections

Page No

I have remaining concerns about how participants were recruited,

Concerns regarding the participant’s recruitment were addressed.

4

Still am not convinced that a study of "nurse learners" is broadly applicable, or more specifically, that it is a good fit with the current journal.

The learner nurse was replaced by “students” for the study to be broadly applicable to the journal

All pages

The complete replacement of the original theoretical framework with an entirely new framework is very concerning and suggests that the study was not theoretically grounded in the first place. 

The framework was replaced based on the suggestions from the first reviewer.

3